# Segmentation-Assisted Fusion-Based Classification for Automated CXR Image Analysis

**DOI:** 10.3390/s25154580

**Published:** 2025-07-24

**Authors:** Shilu Kang, Dongfang Li, Jiaxin Xu, Aokun Mei, Hua Huo

**Affiliations:** Information Engineering College, Henan University of Science and Technology, Luoyang 471000, China; ksl_life@163.com (S.K.); lidongfang2022@126.com (D.L.); xujiaxin0523@163.com (J.X.); 210321050459@stu.haust.edu.cn (A.M.)

**Keywords:** chest X-ray, lung diseases, segmentation model, image classification, lightweight models

## Abstract

Accurate classification of chest X-ray (CXR) images is crucial for diagnosing lung diseases in medical imaging. Existing deep learning models for CXR image classification face challenges in distinguishing non-lung features. In this work, we propose a new segmentation-assisted fusion-based classification method. The method involves two stages: first, we use a lightweight segmentation model, Partial Convolutional Segmentation Network (PCSNet) designed based on an encoder–decoder architecture, to accurately obtain lung masks from CXR images. Then, a fusion of the masked CXR image with the original image enables classification using the improved lightweight ShuffleNetV2 model. The proposed method is trained and evaluated on segmentation datasets including the Montgomery County Dataset (MC) and Shenzhen Hospital Dataset (SH), and classification datasets such as Chest X-Ray Images for Pneumonia (CXIP) and COVIDx. Compared with seven segmentation models (U-Net, Attention-Net, SegNet, FPNNet, DANet, DMNet, and SETR), five classification models (ResNet34, ResNet50, DenseNet121, Swin-Transforms, and ShuffleNetV2), and state-of-the-art methods, our PCSNet model achieved high segmentation performance on CXR images. Compared to the state-of-the-art Attention-Net model, the accuracy of PCSNet increased by 0.19% (98.94% vs. 98.75%), and the boundary accuracy improved by 0.3% (97.86% vs. 97.56%), while requiring 62% fewer parameters. For pneumonia classification using the CXIP dataset, the proposed strategy outperforms the current best model by 0.14% in accuracy (98.55% vs. 98.41%). For COVID-19 classification with the COVIDx dataset, the model reached an accuracy of 97.50%, the absolute improvement in accuracy compared to CovXNet was 0.1%, and clinical metrics demonstrate more significant gains: specificity increased from 94.7% to 99.5%. These results highlight the model’s effectiveness in medical image analysis, demonstrating clinically meaningful improvements over state-of-the-art approaches.

## 1. Introduction

Lung diseases pose a great threat to human health globally, such as coronavirus, pneumonia, and tuberculosis [1], which are significant factors to the onset of lethal diseases. Chest X-ray imaging is the first-line imaging modality for suspected pneumonia and tuberculosis diagnosis per WHO guidelines [2]. For COVID-19, the Fleischner Society recommends CXR as a triage tool in resource-limited settings [3]. Chest X-ray imaging [4] is an important tool for diagnosing lung diseases, providing an important reference for doctors by detecting and analyzing the structure and lesions of the lungs. Traditional diagnostic methods mainly rely on clinical examination and physicians’ empirical judgment. However, this method involves a certain degree of subjectivity and the risk of misdiagnosis. With the rapid development of computer vision and artificial intelligence technology, in the classification and recognition of chest X-ray images, computer-aided diagnosis technology is making significant progress [5,6]. Utilizing deep learning algorithms and big data analysis, computers can automatically identify abnormal features in chest X-ray images and assist physicians in making accurate diagnoses. Although some methods have been proposed [7,8,9], we need to focus more on improving the performance to meet the standards required for application in the clinic. Through continuous research and practice, we expect to further enhance the performance of computer-aided diagnostic techniques and make them powerful tools in clinical practice.

In recent years, deep learning approaches have demonstrated significant promise in medical imaging analysis, particularly in the realm of lung disease detection. The key to deep learning methods for disease classification using chest X-rays is feature extraction. Most of the current research focuses on optimizing the network structure to improve the feature extraction capability by improving the depth, width, and connection of the network. For example, the study by the authors of [8] utilized a dual asymmetric architecture based on ResNet and DenseNet to capture more discriminative features on chest X-ray images adaptively. By optimizing the network structure and connectivity, they aimed to improve the model’s ability to recognize different disease features. Huang et al. [10] proposed a method called HRNet, which achieves the effective use of high-resolution features by simultaneously maintaining and propagating information between feature maps of different resolutions. These feature maps are fused at different stages of the network to maintain the quality of the high-resolution features, aiming to perform classification tasks by extracting anomalous features from four different-resolution feature maps. A3Net is a triple-attention network proposed in [11] that utilizes pre-trained DenseNet121 as a backbone network for feature extraction. A3Net enhances the accuracy and stability of image classification tasks by weighting and refining feature maps through three attention modules: channel attention, element attention, and scale attention. Future research can continue to explore more efficient and accurate attention mechanisms to further improve the performance of image classification. In addition, the emergence of attention modules such as ECA [12], CA [13], and CBAM [14] has contributed to enhancing the model’s focus on critical features and improving its representation capabilities across different scales and channels.

In these works, the challenge lies in demanding computations for segmentation and overlooking crucial details in disease classification. To tackle this, streamlined network architectures and lightweight models can help reduce complexity. To address the above problems, we propose a new segmentation-assisted fusion-based classification model. The contributions of this paper include the following:We proposed a two-stage chest X-ray image classification method. This method enables rapid analysis and diagnosis of chest X-ray images. Using our designed method, the lung mask of chest X-ray images can be quickly acquired, which helps to accurately localize lung regions and extract key features. It also enables rapid diagnosis of pneumonia and COVID-19 diseases through image feature fusion.A new lightweight segmentation model, PCSNet (Partial Convolutional Segmentation Network), was designed to accurately acquire lung masks from CXR images.An enhanced lightweight ShuffleNetV2 model was employed for the classification of CXR images. Through the utilization of three attention-gating modules, attention weights for channel, height, and width dimensions are extracted and effectively synthesized to enhance the model’s comprehension of images.

The remaining content of this paper is structured as follows: Section 2 provides a review of related work. Section 3 describes related models. Section 4 outlines the datasets and implementation details. Section 5 presents the experimental results and analysis. Finally, Section 6 concludes the study.

## 2. Related Work

In this section, we will provide a summary of the related work by presenting previous research on chest X-ray images.

In the task of chest X-ray image segmentation, researchers commonly employ deep learning methods, especially convolutional neural networks (CNNs), to achieve accurate segmentation of regions of interest in lung images. U-Net [15], SegNet [16], and Mask R-CNN [17] are some common methods and techniques used for chest X-ray image segmentation.

Liu et al. [18] improved the network using a pre-trained EfficientNet-b4 as the encoder and residual blocks and LeakyReLU activation functions in the decoder within the U-Net network. This network efficiently extracts lung features and avoids gradient instability caused by multiplication operations in gradient backpropagation. Chowdary and Kanhanged [19] proposed a dual-branch network framework comprising an R-I U-Net network for segmenting lung regions and two fine-tuned AlexNet models for feature extraction. These feature vectors are processed by a recursive neural network and finally classified by concatenation. This dual-branch framework design aims to improve classification accuracy and performance. Currently, models used for studying chest X-ray image segmentation face challenges of high computational and parameter requirements. These models typically demand significant computational resources and storage space due to their complex convolutional neural network structures. To tackle this challenge, we devised a lightweight segmentation model named PCSNet, which adopts an encoder–decoder network architecture [16].

The task of classifying chest X-ray images is fundamentally a medical image analysis task. The objective of this task is to classify given chest X-ray images, identifying them as normal or abnormal, or further categorizing them into specific diseases or abnormalities. Guan et al. [20] proposed ConsultNet, a dual-branch architecture designed to learn distinctive features, aiming to simultaneously achieve disease classification and feature selection. Shamrat et al. [21] utilized the fine-tuned MobileLungNetV2 model for analysis. Initially, chest X-ray images underwent preprocessing, including Contrast Limited Adaptive Histogram Equalization (CLAHE) for enhancing image contrast, and Gaussian filtering for noise reduction. Subsequently, the preprocessed images were fed into multiple transfer learning models for classification. Shentu and AI Moubayed [22] proposed an integrated model called CXR-IRGen for generating CXR image reports. The model consists of visual and language modules, where the visual module incorporates a unique prompt design combining text and image embeddings. The language module utilizes large-scale language models and self-supervised learning strategies to improve the generation of CXR reports. Muralidharanr et al. [23] discussed the application of various deep learning models in COVID-19 screening and analyzed their advantages and disadvantages compared to other methods. Additionally, the study investigated the potential value of this imaging method in the management and early treatment of COVID-19 patients. GazeGNN [24] employs a gaze-guided graph neural network that directly integrates raw eye-tracking data with chest X-ray images through a unified graph representation, thereby bypassing computationally intensive visual attention map (VAM) generation. This enables real-time, end-to-end disease classification with enhanced robustness and state-of-the-art performance. Islam et al. [25] propose a segmentation-guided graph neural network; this work replaces human-dependent eye-tracking data with automatically generated lung segmentation masks. Using a lightweight U-Net for segmentation, the system isolates lung regions in chest X-rays, converting these masks into attention maps for graph construction.

In current research, challenges include overfitting due to deep networks, excessive consumption of computational resources, and the neglect of spatial and channel information on diseases.

To address this issue, we ultimately chose to use a lightweight network, ShuffleNetV2 [26], for the final disease classification. Additionally, we devised a new residual block for ShuffleNetV2, incorporating the TripletAttention module [27] to tackle the problem of the model neglecting the spatial and channel information of diseases.

## 3. Model Descriptions

We developed an innovative approach for chest X-ray image segmentation and classification. The flow of chest X-ray image diagnosis is shown in Figure 1. In later sections, details of the model and techniques will be provided.

### 3.1. Lung Region Segmentation

To acquire lung region masks, we developed a PCSNet (Partial Convolutional Segmentation Network) model tailored for precise segmentation of chest X-ray images. As depicted in Figure 2, our PCSNet network was designed with an encoder–decoder architecture and a five-level segmentation framework. The model comprises three essential elements: the encoder, decoder, and connector. The encoder is responsible for feature extraction and encoding of the input image. In our PCSNet, we leverage the initial 13 layers of the VGG16 model as the encoder. This feature extraction process supplies crucial information for the subsequent decoder segment, enhancing the precision and reliability of our network in accurately segmenting chest X-ray images.

The decoder is essential for reconstructing and retrieving encoder features. PCSNet uses pooling index up-sampling and partial convolution [28] to enlarge the feature map and details. It can create a mask output matching the input image size. The decoder’s structure and recovery process help capture feature information at different scales, helping improve image reconstruction and mask generation. With decoder processing, PCSNet is good at chest X-ray segmentation.

As depicted in Figure 3, partial convolution selectively processes input channels during the convolution operation while ignoring information from other channels, resulting in reduced computational load. Given the typically irregular boundaries of lesions in chest X-ray images, the nonlinear nature of partial convolution enables us to better capture such irregularities. In comparison to traditional convolutional approaches, partial convolution excels in preserving edge clarity and accuracy, particularly in scenarios involving irregular boundaries. This precision in segmentation, along with enhanced descriptive capability for boundaries, enhances the overall effectiveness and performance of lung lesion segmentation tasks in X-ray images.

The input feature map denoted as $I\in R_{\left(c\times h\times w\right)}$ undergoes partial convolution with a k×k convolution kernel to generate the output feature map $O\in R_{\left(c\times h\times w\right)}$. When calculating the FLOPs for consecutive or uniform memory accesses, we treat the first or last $c_{p}$ channels consecutively to represent the entire feature map computationally. Assuming an equal number of input and output feature map channels, the FLOPs for partial convolution can be determined using Equation (Equation 1), while memory accesses (MA) can be calculated using Equation (Equation 2).(1)$$FLOPs=h\times w\times k^{2}\times c_{p}^{2}\;$$(2)$$MA=h\times w\times 2c_{p}+k^{2}\times c_{p}^{2}\approx h\times w\times 2c_{p}\;$$

When the partial convolution ratio r ($c_{p}$/c) is 1/4, the FLOPs of the partial convolution is only 1/16 of the regular convolution, while the memory accesses are only 1/4 of the regular convolution. This optimization strategy not only improves the computational efficiency but also effectively reduces the read and write operations of the memory, which further improves the efficiency of the convolution algorithm.

In PCSNet, the connector refers to the skip connection pathways between encoder and decoder stages (indicated by gray arrows in Figure 2), which is used to merge low-level encoder features with high-level decoder features. This fusion enables the integration of features across different levels and scales, empowering the model to effectively interpret and utilize feature information from diverse hierarchical levels. In Figure 4, the approach of pooling indices in PCSNet diverges from the deconvolution method utilized in U-Net. Unlike U-Net’s use of deconvolution for up-sampling by learning a reversible convolution kernel, PCSNet leverages preserved pooling layer indices for up-sampling to restore feature map resolution. This technique effectively preserves boundary information, enhances model sparsity, and eliminates the need to learn numerous inverse convolution parameters. Additionally, instead of simple concatenation, the preserved information is combined with the corresponding encoder feature maps through summation. This summation operation streamlines network training mitigates vanishing gradient issues and has the potential to deliver superior results in chest X-ray image segmentation tasks.

### 3.2. Classification Based on Segmented Lung Region

The chest X-ray image is segmented and the lung mask is isolated using the pre-trained PCSNet model. We perform element-wise summation between the original chest X-ray image and the lung mask to generate enhanced images with more detailed information, as illustrated in the upper section of Figure 1. The resulting enhanced image is subsequently classified using the ShuffleNetV2 [26] network for diagnostic purposes. ShuffleNetV2 is a lightweight network architecture with low computational and storage resource consumption, which is suitable for efficient image diagnosis on devices with limited computational resources.

ShuffleNetV2 can flexibly control channel counts and dimensionality reduction depending on strides. Following each block in the network, the functions of Triplet Attention and Channel Shuffle are performed. The Channel Shuffle operation plays a crucial role in enhancing the richness of feature representation and reducing redundancy among features. Through this process, channels from distinct groups can establish closer relationships. Triplet Attention complements Channel Shuffle by addressing the potential loss or confusion of feature information, enhancing the model’s expressiveness and overall performance.

#### Triplet Attention

As shown in Figure 5, Triplet Attention is a module comprising three branches that take an input tensor and produce a refined tensor. When presented with an input tensor $X\in R_{\left(C\times H\times W\right)}$, it is sequentially transmitted to each of the three branches.

In the first branch, the input tensor ${X\;}_{\left(C\times H\times W\right)}$ is rotated counterclockwise by 90 degrees along the H dimension to yield the rotated tensor $X_{1\;(W\times H\times C)}$. Subsequently, $X_{1}$ undergoes dimension reduction to form a tensor of shape $X_{1\;(2\times H\times C)}^{\prime }$ through Z-pool. Following this reduction, $X_{1}^{\prime }$ is processed by a convolutional layer with a k × k kernel size to generate an output vector with the shape (1×H×C). Attention weights are computed using a sigmoid activation function, and these weights are then applied to $X_{1}$. Finally, the vector $y_{1}\in R_{\;(C\times H\times W)}$ is rotated 90 degrees clockwise to retain the same shape as the input tensor *X*.

In the second branch, the input tensor *X* is rotated 90 degrees counterclockwise along the W dimension to obtain the rotated tensor $X_{2\;(H\times C\times W)}$. Then, $X_{2}$ is reduced to a tensor $X_{2}^{\prime }$ of shape (2×C×W) by a Z-pool. Similarly, $X_{2}^{\prime }$ is passed through a convolutional layer with a k × k kernel size to obtain an output tensor with dimensions (1×C×W). The next operation is similar to that of the first branch to obtain the output vector $y_{2}\in R_{\;(C\times H\times W)}$.

For the third branch, the channels of the input tensor *X* are directly reduced to 2 via Z-pool to yield the tensor $X_{3\;(2\times H\times W)}^{\prime }$. Subsequently, a convolutional layer with a k × k kernel size processes this tensor to produce an output of shape (1×H×W). The attention weights are generated by applying a sigmoid activation function, which is then used to weight the input tensor *X*, resulting in the vector $y_{3}\in R_{\;(C\times H\times W)}$.

Lastly, the feature tensors $y_{1}$, $y_{2}$, and $y_{3}$ are combined using Equation (Equation 3) to derive the final tensor *y*.(3)$$y=\frac{1}{3}\left(y_{1}+y_{2}+y_{3}\right)\;$$

Equation (Equation 4) is derived by integrating the specific execution steps of the three branches into Equation (Equation 3).(4)$$\begin{array}{c} y=\frac{1}{3}(X_{1}\sigma \left(\psi_{1}({X_{1}}^{\prime }\right))+X_{2}\sigma \left(\psi_{2}({X_{2}}^{\prime }\right))+X\sigma \left(\psi_{3}{X_{3}}^{\prime }\right))) \end{array}$$
where $\psi_{1}$, $\psi_{2}$, and $\psi_{3}$ refer to a convolutional layer operation with a k × k kernel within the three branches, and σ represents the sigmoid activation function.

The Z-pool layer is designed to minimize the zeroth dimension of the tensor following average pooling and maximum pooling operations down to 2. Equation (Equation 5) is applied to determine the resultant shape of the tensor after Z-pool, such that for a tensor shaped as (C×H×W), and the output shape following Z-pool operation is (2×H×W).(5)$$Z-pool\left(X\right)=[MaxPool_{0d}(X),AvgPool_{0d}\left(X)\right]$$

## 4. Experiment

This section covers the experimental setup, including the datasets, parameter settings, training methods, and evaluation metrics.

### 4.1. Dataset

The inclusion of these openly available datasets facilitates a thorough validation and analysis of our proposed model.

**Montgomery County Dataset (MC) [29]**: This dataset was collected from the tuberculosis (TB) control program of the Department of Health and Human Services (HHS) in Montgomery County (MC), Maryland, USA. As shown in Table 1, the Montgomery County Dataset comprises 138 images in total. These were partitioned into training and test sets at an 8:2 ratio, resulting in 111 images for training and 27 images for testing.

**Shenzhen Hospital Dataset (SH) [29]**: The standard digital image database for tuberculosis was created by the National Library of Medicine, Maryland, USA, in collaboration with Shenzhen No. 3 People’s Hospital, Guangdong Medical College, Shenzhen, China. As shown in Table 1, a total of 530 images were utilized for training, with 132 images designated for testing purposes.

**Chest X-Ray Images for Pneumonia (CXIP) [30]**: This dataset was derived from a database comprising chest X-ray scans of children aged one to five years gathered from the Guangzhou Women’s and Children’s Medical Center. The training set has 4192 images, the validation set has 1040 images, and the test set has 624 images, as shown in Table 1. CXIP contains two classes (Normal/Pneumonia) with pneumonia cases confirmed by radiologists. Detailed class distributions are provided in Table 2.

**COVIDx Dataset [31]**: These data were sourced from various countries and clinical facilities, encompassing both adult and pediatric patients afflicted with COVID-19 pneumonia. The dataset amalgamates multiple datasets, including BIMCV [32], RSNA Pneumonia Test [33], RSNA-RICORD [34], SIRM [35], and COHEN [36]. As shown in Table 1, the dataset was partitioned into a training set (26940 X-ray images), validation set (3046 X-ray images), and test set (400 X-ray images). COVIDx contains three classes (Normal/COVID-19/Other Pneumonia), where COVID-19 labels are RT-PCR validated. Detailed class distributions are provided in Table 2.

The selection of these datasets was based on their diversity, representativeness, and relevance to the investigation of tuberculosis, pneumonia, and COVID-19 cases. The experimental results presented in Section 5 were obtained on the independent test sets of each dataset.

### 4.2. Implementation Details

We developed segmentation and classification models using PyTorch (version: 1.10.2; creator: FAIR; location: Menlo Park, CA, USA) on a system with an 4090 GPU (NVIDIA, Santa Clara, CA, USA).

The PCSNet segmentation model was trained on the MC and SH datasets for 200 epochs with a batch size of 16, using a learning rate of 0.0004 and the BCEWithLogitsLoss function. Data augmentation included resizing images to 256 × 256 pixels and applying rotation and flips.

For classification on CXIP and COVIDx datasets, images were resized to 256 × 256 pixels. The model was trained for 40 epochs with a batch size of 64, a learning rate of 0.003, and the CrossEntropyLoss function. The ADAM optimizer was utilized for training.

### 4.3. Evaluation Metrics

In this study, we use several evaluation metrics to assess the performance of our proposed model. These evaluation metrics include accuracy (ACC), recall (RECALL), Dice coefficient (DICE), and Jaccard score (JS). The calculations are as follows:(6)$$\mathrm{Accuracy}\;\left(\mathrm{ACC}\right)\;=\;\frac{\mathrm{TP}\;+\;\mathrm{TN}}{\mathrm{TP}\;+\;\mathrm{TN}\;+\;\mathrm{FP}\;+\;\mathrm{FN}}$$(7)$$\mathrm{Recall}\;\left(\mathrm{RECALL}\right)\;=\;\frac{\mathrm{TP}}{\mathrm{TP}\;+\;\mathrm{FN}}$$(8)$$\mathrm{Dice}\;\mathrm{coefficient}\;\left(\mathrm{DICE}\right)\;=\;\frac{2\mathrm{TP}}{2\mathrm{TP}\;+\;\mathrm{FP}\;+\;\mathrm{FN}}$$(9)$$\mathrm{Jaccard}\;\mathrm{score}\;\left(\mathrm{JS}\right)\;=\;\frac{\mathrm{TP}}{\mathrm{FP}\;+\;\mathrm{TP}\;+\;\mathrm{FN}}$$

Common metrics for classification models are accuracy, recall, negative predictive value (NPV), precision, specificity, and F1-score:(10)$$\mathrm{NPV}=\frac{\mathrm{TN}}{\mathrm{TN}+\mathrm{FN}}$$(11)$$\mathrm{Precision}=\frac{\mathrm{TP}}{\mathrm{TP}+\mathrm{FP}}$$(12)$$\mathrm{Specificity}=\frac{\mathrm{TN}}{\mathrm{TN}+\mathrm{FP}}$$(13)$$F1-\mathrm{score}=2\times \frac{\mathrm{Precision}\times \mathrm{Recall}}{\mathrm{Precision}\;+\;\mathrm{Recall}}$$

TP stands for true positive, which is the number of examples that the model correctly classified as positive. TN stands for true negative, which is the number of examples that the model correctly classified as negative. FP stands for false positive, which is the number of examples that the model incorrectly classified as positive. FN stands for false negative, which is the number of examples that the model incorrectly classified as negative.

For the model to be lightweight, we focus on complexity and size, with FLOPs for inference computation and PARAMS for parameters, optimizing them for a more lightweight model for better efficiency and practicality considering deployment resources.

## 5. Results and Analysis

This section thoroughly examines experimental outcomes for segmentation and classification models through ablation experiments. A comparison with state-of-the-art methods validates our models’ efficacy, providing strong evidence to assess and enhance performance.

### 5.1. Lung Segmentation

We validate the efficacy and generalization capability of the PCSNet model by analyzing its experimental segmentation results on the MC and SH datasets and comparing them with current state-of-the-art methods. Alongside our PCSNet model, we trained U-Net [15], SegNet [16], Attention-Net [37], FPNNet [38], DANet [39], DMNet [40], and SERT [41] models, evaluating them on the MC dataset for validation. The experimental results summarized in Table 3 reveal that our proposed model achieves high accuracy and exceptional segmentation performance on the MC dataset, with efficient computational resource utilization. As depicted in Table 3, we also evaluated the PCSNet model on the SH dataset. It shows great performance in metrics, confirming its stability and adaptability.

### 5.2. Segmentation Method Analysis

Based on the data presented in Table 3, Ours stands out compared to the state-of-the-art techniques. On the MC dataset, our model achieves the highest levels of ACC at 0.9894, DICE at 0.9786, recall at 0.9768, and JS at 0.9582. Additionally, on the SH dataset, our method delivers robust performance with an accuracy of 0.9873, a Dice coefficient of 0.9740, a recall of 0.9701, and a Jaccard score of 0.9494. In summary, our approach demonstrates superior levels of accuracy, Dice coefficient, recall, and JS metrics on both the MC and SH datasets, thereby confirming the efficacy and generalizability of the model we developed compared to existing state-of-the-art methods.

Figure 6 exhibits sample test images capturing lung segmentation outcomes produced by actual masks and various trained models including U-Net, SegNet, Attention-Net, FPNNet, DANet, DMNet, SERT, and Our model. These images highlight the performance disparities among the different models in lung segmentation. It is worth noting that quantitative evaluation of the segmentation performance on the classification dataset is not feasible, as there is a lack of genuine masks for this dataset. Hence, only a qualitative assessment can be conducted to ascertain the accuracy of chest X-ray image segmentation.

As shown in Table 4, we further explored the specific impacts of different components on the MC dataset while keeping other settings unchanged. PConv outperformed Conv in all aspects. Combining the Add and pooling index methods led to better results compared to using them individually. Moreover, the FLOPS and PARAMS were similar.

### 5.3. Lung Disease Classification

We trained classification models using the CXIP and COVIDx datasets. Our model is general and capable of integrating with multiple backbone networks. We conducted comparative experiments using diverse backbone networks to identify the optimal backbone for this study, including ResNet34, ResNet50, DenseNet121, SwinTransformers, and ShuffleNetV2. By training on these diverse models, we aimed to compare their performance on the CXIP and COVIDx datasets to identify the most suitable model for our study. As summarized in Table 5, we assessed the performance of several models on the dataset. On the COVIDx dataset, we also evaluated the performance of Ours. This still maintained high levels of performance. This suggests that Ours possesses a degree of generalization capability, enabling improved identification of positive and negative image samples within the chest X-ray dataset.

Without-Seg means using the original image for training during the training process. With-seg means using the migration learning technique to obtain the lung mask image with PCSNet, and then merging the mask image with the original image for training. Figure 7 illustrates some instances of lung mask images with suboptimal results, indicating that training with With-Seg may adversely impact the classification outcomes. To address this challenge, we employ fusion (ADD) for training. The fusion approach entails training the model on the original images after integrating features extracted from the masked image. In comparison to Without-Seg, training with With-Seg yields enhanced performance on the CXIP and COVIDx datasets by mitigating the effects of noise interference. Moreover, fusion serves to further enhance the classification efficacy by amalgamating information from the original image and the masked image through feature fusion.

### 5.4. Classification Method Analysis

Table 6 presents the experimental findings of this study, comparing various methods in pneumonia classification on the CXIP dataset, with our approach showcasing outstanding performance. In contrast to alternative methods, Ours exhibits notable advantages across key metrics including accuracy, negative predictive value, precision, recall, specificity, and F1-score. In particular, our method achieves 0.9855 in accuracy, which is superior to the 0.9841 of AAPSO [42], 0.9620 of Cropped ROI [43], 0.9836 of [44], 0.9794 of VGG19-RF [45], and 0.9810 of CovXNet [46]. Furthermore, our method demonstrates strong performance with a negative predictive value of 0.9769, precision of 0.9886, recall of 0.9915, specificity of 0.9692, and F1-score of 0.9901. Table 7 offers a comprehensive analysis of various methods in the COVID-19 classification task using the COVIDx dataset. Notably, our approach excels in COVID-19 classification compared to alternative methods, showcasing exceptional performance. Our method particularly shines in the accuracy metric, attaining a commendable score of 0.9750, indicating precise identification of COVID-19 samples. Moreover, our model delivers impressive results in metrics such as precision, recall, and F1-score. These findings underscore the robustness and strong generalization capabilities of our model, validating its efficacy in COVID-19 classification.

### 5.5. Ablation Experiments

To validate the efficacy of the employed methodologies, we conducted image classification experiments on the CXIP dataset, exploring various combinations of three key methods: With-Seg, TripltAttention, and fusion (ADD). As delineated in Table 8, the utilization of either With-Seg or Triplet Attention methods independently resulted in enhanced evaluation metrics compared to their non-utilization. These results demonstrate that PCSNet achieves precise lung mask segmentation and our proposed segmentation model significantly enhances the performance of the classification model. Triplet Attention applies channel–height–width weighting to prevent feature degradation during propagation. Moreover, upon integrating the fusion method into the image classification process, a notable enhancement in effectiveness was observed. The fusion method serves to further enhance the classification efficacy by amalgamating global context from original images with lesion-specific features from masked images. Notably, the optimal image classification outcomes were attained through the simultaneous utilization of all three methods. This comprehensive approach yielded commendable metrics, including an accuracy of 0.9855, precision of 0.9886, recall of 0.9915, and F1-score of 0.9901. The detailed analysis of these results underscores the effectiveness of our methodology.

### 5.6. Visualization Analysis

Ensuring the network’s proficiency in assimilating feature information from medical images is crucial for enhancing model interpretability. The utilization of the Grad-CAM [51] technique serves to generate a heat map of the chest X-ray image, enabling the visualization of the specific regions of focus for the model. Figure 8 showcases the classification results based on the fused images, underscoring the practical viability of our proposed methodology. This approach holds promise for practical implementation and proves pivotal in addressing noise interference challenges in chest X-ray image classification, ultimately enhancing overall model performance. In our model, the final decision is determined by the highlighted region in Figure 8. The model identifies high-attention areas in a heat map that exhibit significant overlap with critical anatomical structures (e.g., pulmonary consolidation zones, pleural adhesion regions). These high-attention regions strongly correlate with key pathological features, substantially enhancing the interpretability of model predictions. For medical professionals, these highlighted areas in the image along with the predicted outcomes serve as valuable aids in disease diagnosis.

## 6. Discussion

This paper proposes a lightweight CXR analysis method—the segmentation-assisted fusion-based classification framework. It utilizes PCSNet for image segmentation, followed by the classification of fused X-ray images using enhanced ShuffleNetV2. Experimental results demonstrate that our approach achieves exceptional performance with minimal computational resource consumption. While recent studies such as HRNet [10], A3Net [11], and DANet [39] have improved classification accuracy, they neglect computational efficiency. Our work achieves a balance between computational efficiency and recognition accuracy. To the fusion strategy, we introduce fundamental innovations: distinct from mainstream dual-branch feature fusion methods [19], our solution integrates lung masks with raw CXR images.

Current CXR analysis faces core challenges including difficulties in collecting high-quality samples and low clinical trust. Moving forward, we plan to delve deeper into multi-label disease classification, a classification task with broad applications in the medical field. Our goal is to optimize the model structure, enhance feature expression, and improve the accuracy and robustness of the models by exploring correlations among different diseases. Meanwhile, few-shot learning and self-supervised learning can effectively reduce models’ dependency on large-scale datasets. Integrating these techniques with CXR analysis represents a critical direction for future research.

## 7. Conclusions

In this paper, a two-stage segmentation-assisted fusion-based classification approach is presented, encompassing segmentation via PCSNet and classification with enhanced ShuffleNetV2. The accuracy, Dice coefficient, recall, and intersection and concurrency ratio for the segmentation of chest X-ray images were 98.94%, 97.86%, 97.68%, and 95.82%, respectively. Then, three classification experiments were conducted: using original X-ray images (Without-seg), segmented lung images (With-seg), and fused X-ray images (fusion). Results showed improved metrics for fusion compared to With-seg and Without-seg: accuracy (98.55% vs. 97.78%, 97.18%), NPV (97.69% vs. 95.60, 95.21), precision (98.86% vs. 98.59%, 97.90%), recall (99.15% vs. 98.36%, 98.24%), specificity (96.62% vs. 96.20%, 94.30%), and F1-score (99.01% vs. 98.48%, 98.07%). The fusion method yielded the best results for COVID-19 classification: accuracy of 97.50%, NPV of 95.67%, precision of 99.48%, recall of 95.50%, specificity of 99.50%, and F1-score of 97.45%. This study used the Grad-Cam method to create heat maps for precise diagnosis, aiding radiologists in swift patient screening and reducing decision-making wait times.

## Figures and Tables

**Figure 1 sensors-25-04580-f001:**
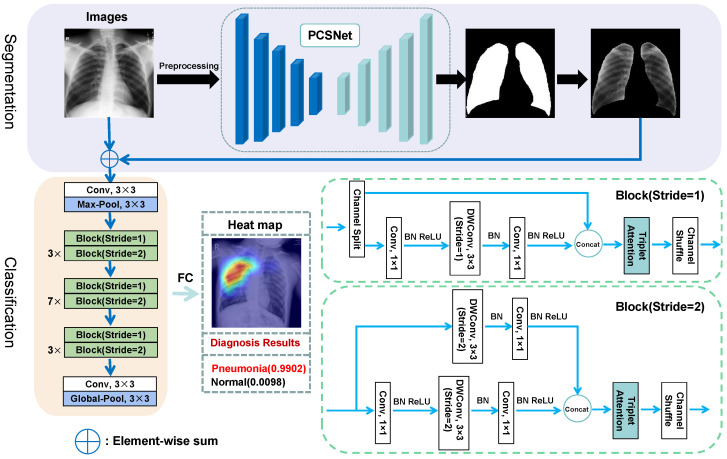
Flow of chest X-ray image diagnosis. The two blocks on the right represent the network implementation details of Block (Stride = 1) and Block (Stride = 2).

**Figure 2 sensors-25-04580-f002:**
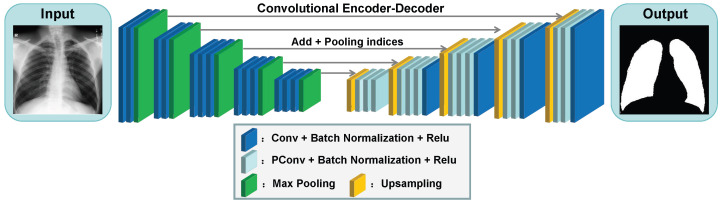
Architecture of PCSNet network with encoder–decoder and five-level segmentation framework.

**Figure 3 sensors-25-04580-f003:**
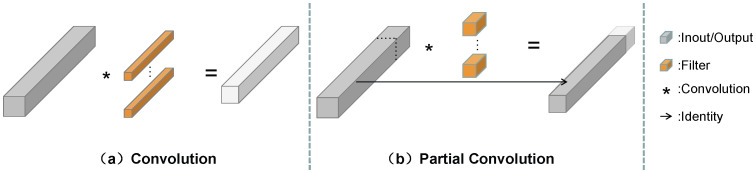
Partial convolution (PConv).

**Figure 4 sensors-25-04580-f004:**
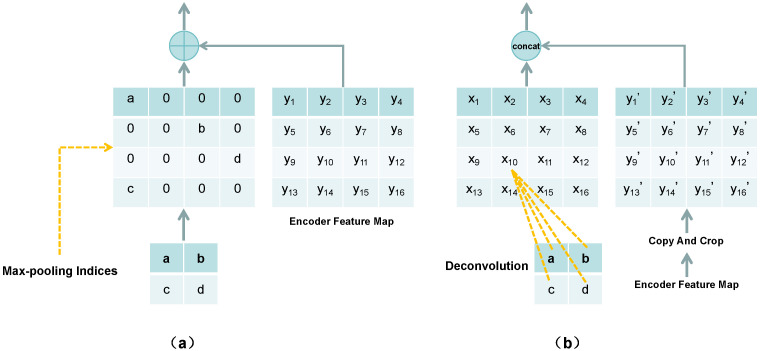
Comparison of pooling index approaches (**a**) in PCSNet and deconvolution method (**b**).

**Figure 5 sensors-25-04580-f005:**
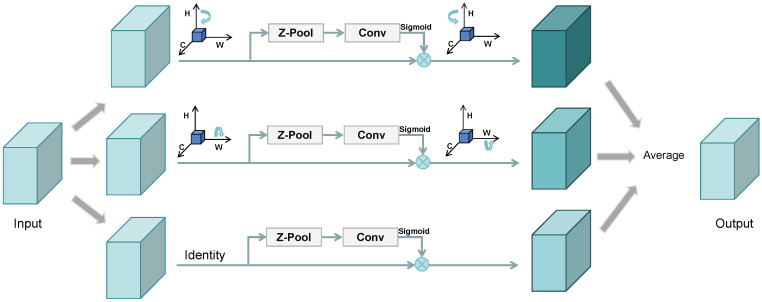
Triplet Attention with three branches.

**Figure 6 sensors-25-04580-f006:**
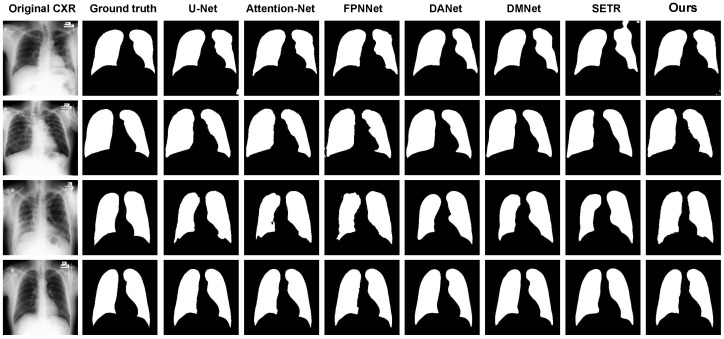
Segmentation outcome comparison of lung images using U-Net, SegNet, Attention-Net, FPNNet, DANet, DMNet, SERT, and Our model.

**Figure 7 sensors-25-04580-f007:**
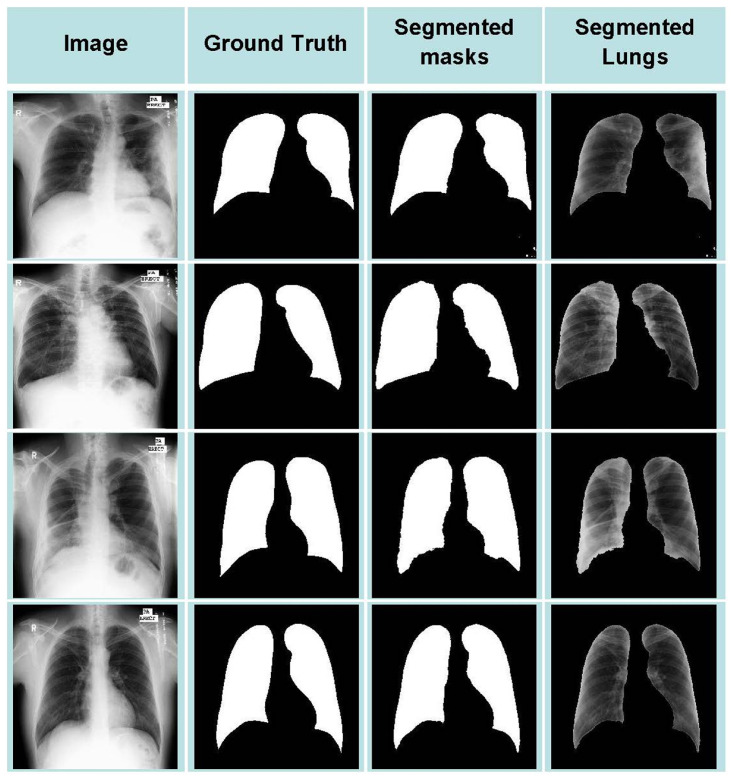
Instances of lung mask images with suboptimal results in CXIP dataset.

**Figure 8 sensors-25-04580-f008:**
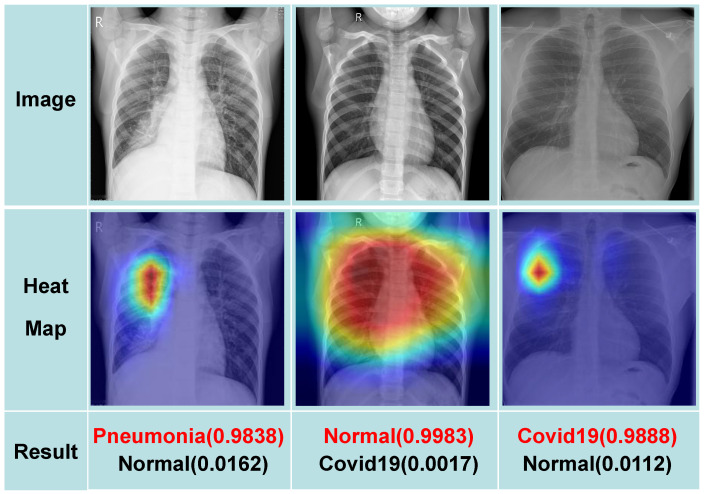
Fused image classification results show practical viability and enhanced model performance, with additional insights from the Grad-CAM generated heatmap.

**Table 1 sensors-25-04580-t001:** Distribution of image counts across training and test sets for MC, SH, CXIP, and COVIDx datasets.

Dataset	MC	SH	CXIP	COVIDx
train	111	530	4192	26,940
validation	-	-	1040	3046
test	27	132	624	400
total	138	662	5856	30,386

**Table 2 sensors-25-04580-t002:** Class distribution in classification datasets.

Dataset	Class	Train	Validation	Test	Total
CXIP	Normal	1082	267	234	1583
	Pneumonia	3110	773	390	4273
COVIDx	Normal	7584	853	100	8537
	COVID-19	14,374	1620	200	16,194
	Other pneumonia	4982	573	100	5655

**Table 3 sensors-25-04580-t003:** Experimental segmentation result comparison on datasets with various models, including U-Net, SegNet, Attention-Net, FPNNet, DANet, DMNet, SERT, and PCSNet. Evaluation based on accuracy, Dice coefficient, recall, Jaccard score, FLOPS, PARAMS, and GPU memory.

Dataset	Model	ACC	DICE	RECALL	JS	FLOPS (G)	PARAMS (M)	GPU Memory (G)
MC	U-Net	0.9836	0.9731	0.9702	0.9477	54.74	31.04	6.74
	Attention-Net	0.9875	0.9756	0.9692	0.9525	135.34	57.16	8.73
	SegNet	0.9850	0.9710	0.9692	0.9437	40.13	29.44	5.70
	FPNNet	0.9837	0.9672	0.9691	0.9365	50.38	34.72	9.28
	DANet	0.9830	0.9668	0.9626	0.9358	125.14	46.19	11.77
	DMNet	0.9821	0.9647	0.9605	0.9318	54.19	51.84	11.99
	SETR	0.9764	0.9559	0.9436	0.9109	8.65	64.00	6.29
	Ours	0.9894	0.9786	0.9768	0.9582	22.19	15.92	5.38
SH	Ours	0.9873	0.9740	0.9701	0.9494	22.19	15.92	5.38

**Table 4 sensors-25-04580-t004:** Exploration of specific impacts of different components on the MC dataset.

Method	ACC	DICE	RECALL	JS	FLOPS (G)	PARAMS (M)	Inference Time (s)
Conv	0.9865	0.9738	0.9709	0.9493	40.13	29.44	0.0256
PConv (Ours)	0.9894	0.9786	0.9768	0.9582	22.19	15.92	0.0253
Add	0.9833	0.9711	0.9712	0.9437	22.19	15.92	0.0255
Pooling indices	0.9840	0.9711	0.9695	0.9439	22.19	15.92	0.0263
Add + pooling indices (Ours)	0.9894	0.9786	0.9768	0.9582	22.19	15.92	0.0253

**Table 5 sensors-25-04580-t005:** Experimental classification result comparison on datasets with various models.

		Without-Seg						With-Seg						Fusion (AFF)					
Dataset	Method	ACC	NPV	PRECISION	RECALL	SPECIFICITY	F1-Score	ACC	NPV	PRECISION	RECALL	SPECIFICITY	F1-Score	ACC	NPV	PRECISION	RECALL	SPECIFICITY	F1-Score
CXIP	ResNet34	0.9667	0.9541	0.9711	0.9836	0.9209	0.9773	0.9709	0.9434	0.9812	0.9789	0.9494	0.9801	0.9720	0.9610	0.9760	0.9860	0.9345	0.9809
	ResNet50	0.9547	0.9122	0.9706	0.9672	0.9209	0.9689	0.9598	0.9243	0.9730	0.9719	0.9272	0.9725	0.9616	0.9401	0.9693	0.9784	0.9163	0.9738
	DenseNet121	0.9658	0.9340	0.9777	0.9754	0.9399	0.9766	0.9718	0.9579	0.9768	0.9848	0.9367	0.9808	0.9748	0.9607	0.9800	0.9857	0.9455	0.9828
	SwinTransforms	0.9530	0.8850	0.9807	0.9543	0.9494	0.9674	0.9701	0.9577	0.9745	0.9848	0.9304	0.9796	0.9718	0.9436	0.9824	0.9789	0.9525	0.9806
	ShufflenetV2	0.9709	0.9638	0.9734	0.9871	0.9272	0.9802	0.9744	0.9413	0.9870	0.9778	0.9652	0.9824	0.9803	0.9592	0.9882	0.9848	0.9684	0.9865
	Ours	0.9718	0.9521	0.9790	0.9824	0.9430	0.9807	0.9778	0.9560	0.9859	0.9836	0.9620	0.9848	0.9855	0.9769	0.9886	0.9915	0.9692	0.9901
COVIDx	Ours	0.9225	0.8858	0.9669	0.8750	0.9700	0.9186	0.9735	0.9688	0.9776	0.9726	0.9746	0.9751	0.9750	0.9567	0.9948	0.9550	0.9950	0.9745

**Table 6 sensors-25-04580-t006:** Comparison of pneumonia classification performance on the CXIP dataset.

Method	Class	ACC	NPV	PRECISION	RECALL	SPECIFICITY	F1-Score
AAPSO [42]	Normal/Pneumonia	0.9841	-	0.9880	0.9902	-	0.9891
Cropped ROI [43]	Normal/Pneumonia	0.9620	-	0.9770	0.9620	0.9620	0.9510
CNN [44]	Normal/Pneumonia	0.9836	-	0.9898	0.9879	-	0.9888
VGG19-RF [45]	Normal/Pneumonia	0.9794	-	0.9502	0.9755	-	0.9627
CovXNet [46]	Normal/Pneumonia	0.9810	-	0.9800	0.9850	0.9790	0.9830
Ours	Normal/Pneumonia	0.9855	0.9769	0.9886	0.9915	0.9692	0.9901

**Table 7 sensors-25-04580-t007:** Comparison of COVID-19 classification performance on the COVIDx dataset.

Method	Class	ACC	NPV	PRECISION	RECALL	SPECIFICITY	F1-Score
CovXNet [46]	Normal/COVID	0.9740	-	0.9630	0.9780	0.9470	0.9710
CNN [47]	Normal/COVID	0.9671	-	0.9425	0.9278	-	0.9351
CycleGAN [48]	Normal/COVID	0.9420	-	0.9548	0.9620	-	0.9584
DeTraC deep CNN [49]	Normal/COVID	0.9310	-	-	1.0000	-	-
QCovSML [50]	Normal/COVID	0.9334	-	0.9202	0.9559	-	0.9373
Ours	Normal/COVID	0.9750	0.9567	0.9948	0.9550	0.9950	0.9745

**Table 8 sensors-25-04580-t008:** Analysis of experimental results on the CXIP dataset involving different combinations of experiments to evaluate the effectiveness of the With-Seg, TripltAttention, and fusion (ADD) methods.

With-Seg	Triplt Attention	Fusion (ADD)	ACC	NPV	PRECISION	RECALL	SPECIFICITY	F1-Score
✘	✘	✘	0.9709	0.9638	0.9734	0.9871	0.9272	0.9802
✔	✘	✘	0.9744	0.9413	0.9870	0.9778	0.9652	0.9824
✘	✔	✘	0.9718	0.9521	0.9790	0.9824	0.9430	0.9807
✔	✘	✔	0.9803	0.9592	0.9882	0.9848	0.9684	0.9865
✔	✔	✘	0.9778	0.9560	0.9859	0.9836	0.9620	0.9848
✔	✔	✔	0.9855	0.9769	0.9886	0.9915	0.9692	0.9901

## Data Availability

We clarify that our research findings are based on the analysis of publicly available datasets: Montgomery County Dataset (MC) and Shenzhen Hospital Dataset (SH): https://www.kaggle.com/datasets/felipemeganha/chest-xray-masks-and-labels-images (accessed on 22 March 2025). Chest X-Ray Images for Pneumonia (CXIP): https://www.kaggle.com/datasets/paultimothymooney/chest-xray-pneumonia (accessed on 25 March 2025). COVIDx: https://www.kaggle.com/datasets/andyczhao/COVIDx-cxr2 (accessed on 25 March 2025).

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
