# Peer review of "Segmentation-Assisted Fusion-Based Classification for Automated CXR Image Analysis"

_sensors, 2025, doi:10.3390/s25154580_

Round 1

Reviewer 1 Report

Comments and Suggestions for Authors

In this manuscript, the authors propose a two-stage AI method to first segment the lung region in CXR images and then classify lung diseases. They developed a new model PCSNet, based on partial convolution layers, for segmentation and compared its performance to the one obtained with other state-of-the-art segmentation deep models on public datasets. For the classification step, a Triplet Attention module is inserted into the ShuffleNetV2 classification network. Also in this case, a comparative study to compare the performance of the authors’ approach to other existing models is performed, together with an ablation study. With their approach, the authors obtained higher metrics with respect to the other existing models. However, I have some concerns that should be addressed by the authors to improve the quality of the manuscript.

Major: The article lacks support for the claim that diagnoses of the chosen diseases are actually made in clinical practice using CXR images. As a result, the benefit of their approach is not clearly demonstrated. For instance, CXR images are generally used to define the severity of a pathology rather than to make a definitive diagnosis. If their approach relies on CXRs for diagnosis, they should cite references to support this in their specific context. I also have some reservations about the relevance of the results obtained with their models, as they are only marginally superior to the metrics of the other models they compared. The sole advantage appears to be in the reduced computational power required for their model, but this is not sufficiently emphasized in the text. Additionally, sharing their code would be a valuable addition to their claims. I encourage the authors to make it available.

Minor: The following minor concerns should be addressed before considering the manuscript for publication.

  1. The sentence between lines 2 and 3 of the Abstract is not so clear. What do the authors mean by non-lung features? Is it referring to the segmentation task?
  2. I suggest that the authors specify in the abstract which type of classification they want to solve with their model.
  3. Some references are not very appropriate and related to the specific topic, such as reference number 3. I suggest that the authors check the references in order to only add the more pertinent and updated ones.
  4. In the Related Work section, lines 101-105, the authors give too many details on their model. In my opinion, this is not useful in this specific section. I suggest that the authors delete these lines or move them to another section.
  5. I suggest that the authors change the title of Section 3 from “Approach” to “Models (or Architecture) description”, or something similar.
  6. In line 171 of the Lung Region Segmentation paragraph, a connector is mentioned. What do the authors mean by this connector? It does not seem mentioned before in the manuscript.
  7. I ask the authors if they could repeat the reference to the model ShuffleNetV2 they use for the classification in line 186 of paragraph 3.2.
  8. In paragraph 4.1, I suggest that the authors to specify the total number of images of the Montgomery County Dataset also within the text, as it is explicit for the other datasets.
  9. In Table 1 or in its caption, please specify that the numbers correspond to the number of images.
  10. For the dataset used for the classification, please specify the description of the two classes and that these labels are available in the considered dataset and also clarify the number of images per class in each dataset.
  11. I suggest that the authors divide the Results and Analysis section into two separate sections, moving the Analysis description first, also adding more details on the analysis part, e.g., how they conduct the training of the models, on which datasets? The model tested before on the MC dataset and then on the SH dataset is trained on both datasets together or just on the MC one? This is not clear from the text.
  12. Please add the references to the used models cited in lines 283-284 of the “Lung Segmentation” paragraph.
  13. For the classification step, I suggest that the authors better specify the different classifications for the two considered datasets, i.e., that the pathology that they want to diagnose is different. Also in this case, please clarify on which dataset you trained the tested algorithms.
  14. It is not clear from the text how the mask image is merged with the original image. Are the pixels of the original images outside the mask region set to zero? Please specify this in the manuscript.
  15. In line 322, “training the original image” is not a correct expression.
  16. Could the authors better explain what the feature fusion consists of? It is not clear from the text.
  17. In Table 4, the results of classification With-Seg are obtained using the segmentation obtained with your model? Please clarify.
  18. The obtained results are, in general, very high. Is it possible to add to the text some training graphs, such as loss curves, to prove that the overfitting is avoided? Have the authors used a training-validation-test split? Is the mentioned test set on which the results are reported a separate and independent set of images, or is it the validation one? Have the authors tried a k-fold cross-validation?
  19. In Figure 7, the images from which dataset are shown? Please specify this in the caption.
  20. It is not clear to me why other state-of-the-art models for classification are mentioned in paragraph 5.2.1and table 5 with respect to the ones used in paragraph 5.2 and table 4.
  21. In paragraph 5.3, the reference to Table 7 is not well written.
  22. What do the authors mean by fused images in section 5.4?

23. The text lacks some considerations about the obtained heat maps. Is the highlighted region significant from a clinical point of view? I kindly ask the authors to add these details. These considerations could be added in a Discussion section, which is now missing from the text. I suggest the authors add this additional section to include general considerations on the obtained results, a broader comparison with the literature, and which are the eventual limitations of their work.

Reviewer 2 Report

Comments and Suggestions for Authors
  1. Figure 1 is the Flow of the proposed diagnosis model. As author said, the proposed model has two stages, including segmetion and classification. But Figure 1 did not clearly show these two stages. 
  2. Moreover, in Figure 1, what's the relationship between the "Heat map" box and the two blocks at the right of it?
  3. Line 346 shows "Tab. ??".

Reviewer 3 Report

Comments and Suggestions for Authors

The paper is interesting. I have the following comments

  1. The ablation section of the paper needs some work. Given the complex architecture, there is a need to shown how each of these components contribute to the overall system. The para references to a table that is not linked. Please address this.
  2. Have you looked at calibration metrics? For validating segmentation part , why not use IoU or mIoU?
  3. One issue is that in tables 5 and 6 for accuracy, the proposed methodology barely beats the baselines. Have you conducted any statistical tests?

Reviewer 4 Report

Comments and Suggestions for Authors

Title. Segmentation-assisted Fusion-based Classification for Automated CXR Image Analysis

Summary.

Computer Aided Diagnostic (CAD) systems can automatically identify abnormalities in chest X-ray images and assist physicians in making accurate diagnosis. Although existing deep learning-based classification techniques have made strides in medical image analysis, there is still room for improvements.

Therefore, this paper proposed to aid a disease classification model by providing supplementary data such as, a segmentation mask, for improved performance. In this way, the model focuses on specific regions in the image, and consequently, becoming less prone to errors. For this purpose, a lightweight segmentation model is developed that can efficiently identify the region-of-interest in images. In addition, existing lightweight classification network i.e., ShuffleNetV2 is implemented. The efficacy of their scheme is confirmed on 4 datasets using various evaluation metrics. Their analysis showed that when the classification model is complemented with a segmentation mask there is up-to 1% performance gain.

In general, the proposed methodology has merit, contribution is significant, and the paper is well organized; however, there are some minor concerns. Please refer the comments below.

Comments.

1. Instead of directly reporting performance metric values in the abstract, please discuss what is the performance difference between proposed and an existing best model.

2. The related work section divides the existing art into two parts: the first part discusses works related to segmentation while the second part discusses classification techniques. However, discussion regarding fusion techniques that combine both, as in this work, is missing. Perhaps, relevant literature can be compiled by searching keywords such as, segmentation aided classification or data aided classification etc. Some example works are cited in [1] and [2], please refer to them.

[1]. Wang et al, “Gazegnn: A gaze-guided graph neural network for chest x-ray classification,” 2024. [https://doi.org/10.48550/arXiv.2305.18221]
[2]. Shovon et al., "Segmentation Aided Multiclass Classification of Lung Disease in Chest X-ray Images using Graph Neural Networks," 2025. [doi: 10.1109/ICOIN63865.2025.10992753]

3. The proposed methodology section is mainly dedicated to the discussion of preliminary details; however, the description of proposed methodology has been overlooked. The core idea of this work is to use the segmentation mask with the classification model; however, no information is provided in Section 3. For example, in line#186~187 it is mentioned "The ShuffleNetV2 network then processes the enhanced image for diagnostic purposes." It is not clear what is this enhanced image? How the classification leverages the segmentation mask information? Please revise Section 3.2. by providing the necessary details.

4. In line#148, please use full form for "info" as "information. Also. in line#377, please use the correct term "heatmaps" instead of "thermograms".

=========== end of my comments ===========

Round 2

Reviewer 1 Report

Comments and Suggestions for Authors

The revisions have improved the manuscript, and several of the initial concerns have been addressed. Nonetheless, some points still require clarification or improvement, and I recommend further revisions before acceptance.

  • In Table 7, the results related to the COVID/Normal classification are reported; however, it is unclear from the manuscript text whether only cases labeled as normal and COVID were used in the model's training. If the cases labeled as “other pneumonia” in the COVIDx dataset have been discarded from the analysis, it should be clearly stated in the text.
  • Even though in the answers to the comments, the authors declare that the segregation of the Experimental Results and Analysis sections has been implemented as suggested in the revision, I cannot see this separation of the two sections in the revised manuscript.
  • I suggest the authors also specify in the text that the reported results are obtained on the independent test set of images. Also, adding the number of images in the validation set in Table 2 could be useful to demonstrate the reliability of the analysis process.
  • The authors’ response to comment 20 is clear; however, I suggest the authors clarify this difference better in the text of the manuscript, too.
  • The authors’ response to comment 23 is right; the heatmaps are generally used to visualize the specific regions of the image on which the model focuses most to give the final prediction. Nevertheless, my concern is not on heatmaps in general, but on the clinical significance of the heatmaps the authors obtain for their model on the considered data. Have the regions evidenced in your heatmaps a clinical significance? Are they useful to interpret the prediction of the model?
  • The Discussion section is typically written before the Conclusion. A possible solution could also be to merge the discussion and conclusion sections.

Reviewer 2 Report

Comments and Suggestions for Authors

Table 5 is not clear.

Author Response

Comment: Table 5 is not clear.

Responese: We sincerely appreciate your suggestion! Given the extensive data in Table 5, ​we have horizontally formatted it​ to enhance readability .

Your  insightful suggestions have played a crucial role in elevating the quality of our paper. Best wishes to you!

Reviewer 3 Report

Comments and Suggestions for Authors

The authors addressed the reviewers comments adequately

Author Response

We sincerely appreciate your hard work and dedication!Your  insightful suggestions have played a crucial role in elevating the quality of our paper. Best wishes to you!